# Know-do gaps for cardiovascular disease care in Cambodia: Evidence on clinician knowledge and delivery of evidence-based prevention actions

Nikkil Sudharsanan[1,2]*, Sarah Wetzel[2], Matthias Nachtnebel[3], Chhun Loun[4], Maly Phy[4], Hero Kol[4], Till Bärnighausen[2]

1 Professorship of Behavioral Science for Disease Prevention and Health Care, Technical University of Munich, Munich, Germany, 2 Heidelberg Institute of Global Health, Heidelberg University, Heidelberg, Germany, 3 KfW, Frankfurt, Germany, 4 Department of Preventive Medicine, Ministry of Health, Phnom Penh, Cambodia

* nikkil.sudharsanan@tum.de

**Data Availability Statement:** Our data are freely available in the Open Science Framework

## Abstract

Cardiovascular diseases (CVD) are the leading cause of death in Cambodia. However, it is unknown whether clinicians in Cambodia provide evidence-based CVD preventive care actions. We address this important gap and provide one of the first assessments of clinical care for CVD prevention in an LMIC context. We determined the proportion of primary care visits by adult patients that resulted in evidence-based CVD preventive care actions, identified which care actions were most frequently missed, and estimated the know-do gap for each clinical action. We used data on 190 direct clinician-patient observations and 337 clinician responses to patient vignettes from 114 public primary care health facilities. Our main outcomes were the proportion of patient consultations and responses to care vignettes where clinicians measured blood pressure, blood glucose, body mass index, and asked questions regarding alcohol, tobacco, physical activity, and diet. There were very large clinical care shortfalls for all CVD care actions. Just 6.4% (95% CI: 3.0%, 13.0%) of patients had their BMI measured, 8.0% (4.6%, 13.6%) their blood pressure measured at least twice, only 4.7% (1.9%, 11.2%) their blood glucose measured. Less than 21% of patients were asked about their physical activity (11.7% [7.0%, 18.9%]), smoking (18.0% [11.8%, 26.5%]), and alcohol-related behaviors (20.2% [13.7%, 28.9%]). We observed the largest know-do gaps for blood glucose and BMI measurements with smaller but important know-do gaps for the other clinical actions. CVD care did not vary across clinician cadre or by years of experience. We find large CVD care delivery gaps in primary-care facilities across Cambodia. Our results suggest that diabetes is being substantially underdiagnosed and that clinicians are losing CVD prevention potential by not identifying individuals who would benefit from behavioral changes. The large overall and know-do gaps suggest that interventions for improving preventive care need to target both clinical knowledge and the bottlenecks between knowledge and care behavior.

Repository through the following link: (https://www.doi.org/10.17605/OSF.IO/SA4J5).

**Funding:** This study was supported by the KfW through funding given to NS and TB. This funding was used to collect the the study data and none of the authors received salary support from this source of funding. KfW were involved in the study as research partners.

**Competing interests:** The authors have read over the PLOS policy and note: funding was given from the KfW Group. There are no patents, products in development or marketed products associated with this research to declare. This does not alter our adherence to PLOS Global Public Health policies on sharing data and materials.

## Introduction

Reducing preventable mortality from cardiovascular diseases (CVD) is a major global health priority [1]. In Cambodia, the site of our study, CVDs are estimated to be the leading cause of death [2]. Major risk factors for CVD are also poorly controlled: 12% of all adults ages 40 + have uncontrolled hypertension [3], 32% of men smoke cigarettes [4], and 40% of all individuals with diabetes are untreated [5]. Improving preventive care will thus be essential for meeting CVD mortality reduction targets. This need is especially pressing in middle-income countries like Cambodia, where rapid population aging is expected to dramatically increase the number of individuals in need of preventive care [6].

Conducting a CVD risk assessment where clinicians screen for hypertension, diabetes, and other key risk factors such as physical inactivity and tobacco use is the first essential clinical action for preventing CVD [7]. To address low health-seeking behavior for preventive care [8], major CVD prevention guidelines—including those in Cambodia—recommend that clinicians opportunistically screen and conduct CVD risk assessments for all adult patients that come to their clinics [7, 9]. Whether such efforts translate into improved patient outcomes, however, depends on whether clinicians actually provide guideline-recommended actions. The low levels of diagnosed hypertension and diabetes in Cambodia and other LMICs suggest that clinicians may not be consistently delivering the care actions specified in evidence-based prevention guidelines. To our knowledge, however, there are no studies that have measured clinician behavior regarding CVD prevention care in an LMIC context like Cambodia.

In addition to measuring shortfalls in clinical care for CVD, it is important to identify whether gaps are due to a lack of clinical knowledge or a gap between knowledge and care behavior. Clinicians may simply not be aware of screening guidelines or what actions they should take. This hypothesis is consistent with emerging evidence across Asian countries that finds low levels of clinician care knowledge for common conditions such as diabetes, tuberculosis, and child diarrhea [10–12]. Alternatively, physicians may know that they should conduct opportunistic screenings, but not do so in actual clinical care (this discrepancy between clinicians' knowledge and behavior is referred to as the "know-do" gap [11, 13, 14]). Distinguishing between these two sources is important for crafting effective policy, as interventions to address poor clinical knowledge likely differ substantially from those designed to encourage clinicians with accurate knowledge of evidence-based care to change their care behaviors.

Here we provide one of the first assessments of clinical care for CVD prevention in an LMIC context. Our analysis benefits from data on observations of real patient consultations and clinicians' responses to hypothetical patient vignettes. This allows us to investigate whether care gaps are driven by low levels of clinical knowledge (based on responses to the vignettes) or by a barrier between knowledge and behavior (the difference between responses to the vignettes and care for real patients). We also investigate if clinician knowledge and behavior are related to clinician cadre and overall clinician experience. Overall, our study results are important for assessing the quality of CVD care in Cambodia and informing interventions to improve clinician delivery of evidence-based CVD prevention care.

## Materials and methods

### Setting

Our study takes place in public Health Centres across Cambodia. Health Centres are responsible for delivering primary care throughout the country. The Health Centres provide a range of basic preventive and curative services including care for infectious and acute conditions (e.g.

malaria, respiratory disease, and tuberculosis), family planning and ante and postnatal care, and basic care and screening for chronic non-communicable diseases including screening for hypertension and diabetes. The full range of services that the Health Centres provide is specified in the Cambodian Minimum Package of Activities [15]. Care in the Health Centres is free for poorer families through Cambodia's Health Equity Fund [16]; families that are not eligible for the Health Equity Fund have to pay for their care based on a pre-specified fee schedule [16].

Clinical care in the Health Centres is provided by nurses, midwives, and medical doctors (physicians), with nurses and midwives providing the majority of care. While some of the care responsibilities are differentiated by clinician cadre (e.g. midwives focus more heavily on ante and postnatal care while physicians focus more heavily on the outpatient department), there is substantial overlap in the care responsibilities by cadre, and importantly, all three clinician types are expected to provide CVD preventive care that includes screening for hypertension and diabetes and assessing a range of behavioral risk factors.

Patients that arrive at the Health Centres are first directed to a reception area, after which they are directed to either an outpatient department for general medicine (primary care) or to a maternal child health/delivery department. After their consultation, patients are then directed to the station for wound dressing (if needed following minor surgeries) and pharmacy. If no additional care is required at that time the patient is then sent home; otherwise, the patient is admitted and referred to a public Referral Hospital [15].

The Government of Cambodia established evidence-based care guidelines for cardiovascular disease prevention (in the form of guidelines for nutrition, physical activity, hypertension, and diabetes care) as part of the first National Plan for Prevention and Control of Non-Communicable Diseases 2007–2010. These guidelines were updated for the second, 2013–2020, and third, 2018–2027 plans [17, 18]. Unfortunately, there is no definitive evidence on how thoroughly clinicians in Health Centres were trained on these guidelines; this is an important point we return to when contextualizing our findings in the Discussion section.

## Data

We use data collected from 114 public primary Health Centres across Cambodia between July and September 2020. These data were collected as part of an ongoing evaluation of a country-wide non-communicable disease quality improvement effort. All the study data were collected before the program implementation began.

We used a non-random procedure to select health facilities that was designed to enable an evaluation of the larger quality-improvement intervention. The Government of Cambodia selected 20 out of Cambodia's 94 operational districts to implement the intervention. Each operational district contains on average 13 primary Health Centres which cater to a population between 100,000–200,000 individuals. Within each operational district, the government selected between 2–3 Health Centres to receive the non-communicable disease intervention. We collected data from the 2–3 Health Centres selected by the Government and a control set of 2–3 facilities, chosen to be similar to the intervention facilities in terms of geography and patient caseloads. The median number of patients seen in the facilities per month was 773, ranging from a minimum of 114 to a maximum of 2820 patients.

## Measurements

We collected two sources of data within each selected Health Centre: (1) direct observations of patient-clinician consultations and (2) clinical vignettes. We use the standard definition of "clinician" to mean all individuals who provide clinical patient care. In the Health Centres in

Cambodia, this includes midwives, nurses, and physicians/medical doctors. We collected data from approximately 2 clinical consultations of patients ages 40+ per facility for a total sample of 190 patient observations. To do so, we stationed an enumerator in the clinician's consultation room (after obtaining consent from both the patient and clinician). The observer had a checklist of clinician action items and marked items corresponding to what they observed during the consultation. Importantly, the observer did not directly interact with the patient or the clinician in any way. We observed a unique set of patients in each facility (e.g. it was not the case that the same patient met a nurse, midwife, and physician and was counted as three observations).

We complemented the direct observations with data on clinician responses to a hypothetical patient vignette. Administering clinical vignettes involves providing clinicians descriptions of hypothetical patients and asking them how they would proceed with care. Vignettes are often used as a measure of clinician knowledge and are included as part of many large health facility surveys, including in the Indonesian Family Life Survey [19]. For our study, enumerators presented clinicians with a short description of a hypothetical patient and asked how they would proceed with care. Enumerators described the patient as a 40-year-old who came to the clinic complaining of lower back pain. Clinicians were then asked what questions they would ask the patient, what tests they would run, what medicines or treatments they would prescribe, and what advice or counseling they would provide. Clinicians were not prompted beyond these questions and were only provided responses to the questions or tests they stated they would ask or conduct. We structured the vignette such that the hypothetical patient had uncontrolled blood pressure (values of 160/90 mmHg, 159/92 mmHg, and 159/93 mmHg per measurement respectively), uncontrolled blood glucose (fasting plasma glucose > 126 mg/dl), and a body mass index of 29.4 kg/m^2. The hypothetical patient was also a non-smoker, with no reported alcohol habit, and no reported regular exercise. If the clinician asked a question or stated that they would conduct a test that we did not have pre-determined answers for, enumerators told the clinician that "the patient is unsure or does not know" or "the test came back normal." The vignette was administered to approximately 3 clinicians per facility, for a total sample of 337 vignette responses.

We collected data on the following CVD care items. Measurement items included whether the clinician measured BP at all, whether they measured BP at least twice, whether they measured blood glucose, and whether they measured body mass index. Risk assessment items included whether they asked about the patients' smoking, alcohol, diet, and physical activity behaviors. We measured both whether clinicians conducted these actions with real patients and whether they stated that they would ask/measure these items in their responses to the hypothetical patient vignettes.

We also used data on clinician cadre (whether the clinician was a nurse, midwife, or medical doctor) and the number of years that the clinician reported practicing medicine.

## Statistical analyses

We first estimated the proportion of direct observations where clinicians measured or asked about each CVD care item. We then estimated these same proportions among clinical vignettes and constructed the "know-do" gap as the difference in each of the items between the patient observations and clinical vignettes.

Next, we examined whether there were differences in CVD care across clinician cadre by estimating Poisson regression models for each item with indicator variables for midwives and nurses (with physicians as the reference category). We examined the association with years of experience using similar regression models, this time with a continuous measure of years

practicing medicine as the independent variable. We chose to estimate Poisson, rather than logistic, regression models so that our coefficients can be interpreted as prevalence, rather than odds, ratios. All our standard errors were clustered at the health facility level.

This study received ethics approval from the Ethics Commission of the Medical Faculty of Heidelberg University and the Cambodian National Ethics Committee for Health Research. We obtained written informed consent from health administrators to collect data within health centres and verbal informed consent from clinicians and patients before collecting any data. We used verbal, rather than written consent, to avoid disrupting regular clinical services and to avoid literacy and comprehension challenges with patients. Participant consent was noted by the observer in the data collection form. This procedure was approved by the ethics committees. We conducted all analyses using R Version 4.1.0.

## Inclusivity in global research

Additional information regarding the ethical, cultural, and scientific considerations specific to inclusivity in global research is included in the S1 Checklist.

## Results

Nurses and midwives provided the most medical care (**Table 1**): 152 (45%) of the vignette sample and 33 (17%) of patient observations were midwives, 151 (45%) of the vignette sample and 129 (68%) of the patient observations were nurses, and less than 15% of both samples were medical doctors (vignette sample: 10%, patient observations: 15%). On average, providers had around 11–12 years of experience practicing medicine Vignette sample: 10.8 (9.2), patient observation sample: 12.1 (9.9).

There were very large clinical care gaps for nearly all CVD care actions (**Fig 1**). Just 6.4% (95% CI: 3.0%, 13.0%) of patients had their BMI measured, 8.0% (95% CI: 4.6%, 13.6%) their blood pressure measured at least twice, only 4.7% (95% CI: 1.9%, 11.2%) their blood glucose measured. Similarly, less than 21% of patients were asked about their physical activity (11.7%, 95% CI: 7.0%, 18.9%), smoking (18.0%, 95% CI: 11.8%, 26.5%), and alcohol-related behaviors (20.2%, 95% CI: 13.7%, 28.9%).

Contrasting these results with responses to the clinical vignettes, we observed very large know-do gaps for blood glucose and BMI measurement. Clinicians stated that they would measure blood glucose among 86.6% (95% CI: 82.1%, 90.2%) of vignettes and BMI among 73.3% (95% CI: 68.0%, 78.0%) of vignettes. There were smaller but still important know-do gaps for the remaining care items. Based on vignette responses, 22.6% (95% CI: 17.8%, 28.1%) of clinicians stated they would collect at least two blood pressure measurements, 20.8% (95%

**Table 1. Descriptive characteristics of the sampled clinicians.**

| | Clinicians who responded to the clinical vignettes (N = 337) | Clinicians providing care to the patients whose consultations were observed (N = 190) |
|---|---|---|
| **Qualification of clinicians in each sample** | | |
| medical doctor | 34 (10.1%) | 28 (14.7%) |
| midwife | 152 (45.1%) | 33 (17.4%) |
| nurse | 151 (44.8%) | 129 (67.9%) |
| **Years practicing among clinicians in each sample** | | |
| Mean (SD) | 10.8 (9.22) | 12.1 (9.89) |
| Median [Min, Max] | 8.00 [0, 39.0] | 8.00 [0, 34.0] |

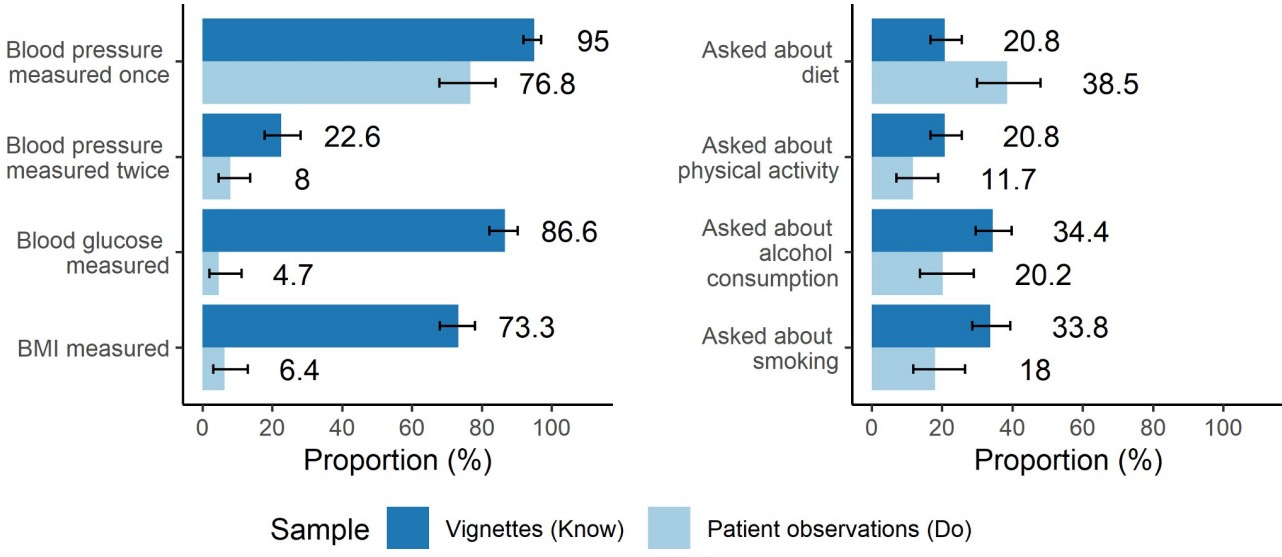

**Fig 1. Know-do gaps for non-communicable disease screening measurement actions and behavioral risk assessments.** Error bars represent 95% confidence intervals.

CI: 16.7%, 25.5%) stated they would ask about physical activity, 33.8% (95% CI: 28.6, 39.4%) stated they would ask about smoking, and 34.4% (95% CI: 29.5%, 39.7%) stated they would ask about alcohol.

We did not find strong evidence of differences in the size of CVD care gaps by clinician cadre (**Fig 2**). The only exception was that midwives (PR: 0.19 [0.056, 0.760]) and nurses (PR: 0.33 [0.122, 0.906]) were less likely to ask patients about their physical activity compared to physicians. We also did not find strong evidence of an association between the number of years a clinician reported practicing medicine and any of the CVD care items (**Fig 3**). The detailed regression results for **Figs 2** and **3** are available in Tables B-Q in S2 Text.

## Discussion

We found evidence of large CVD care delivery gaps by clinicians in public primary-care facilities in Cambodia. Based on direct observations of clinical consultations, virtually no patients had their blood glucose measured and only small shares had their blood pressure measured correctly and were asked about important risk factors. These alarming findings have three important implications. First, due to the low rates of blood sugar measurement, it is likely that many individuals with diabetes are not being detected. Second, clinicians are losing a large and important source of CVD prevention potential by not identifying individuals who would benefit from behavioral changes. Lastly, while many clinicians measured blood pressure once, few measured it at least twice as recommended by most care guidelines [7, 20]. This suggests that clinical decisions around hypertension in Cambodia may be made with substantial error due to potential "white coat effects" [21].

Whether these gaps are driven by a lack of clinical knowledge or a discrepancy between knowledge and action depended on the specific care actions. There were large know-do gaps for blood glucose and BMI measurements, suggesting that knowledge is unlikely to be the main driver of care gaps. We observed smaller know-do gaps for the other CVD care items; for these care behaviors, both improving knowledge and addressing the gap between knowledge and action are likely to be needed to close overall care gaps. To our knowledge, this is the first study

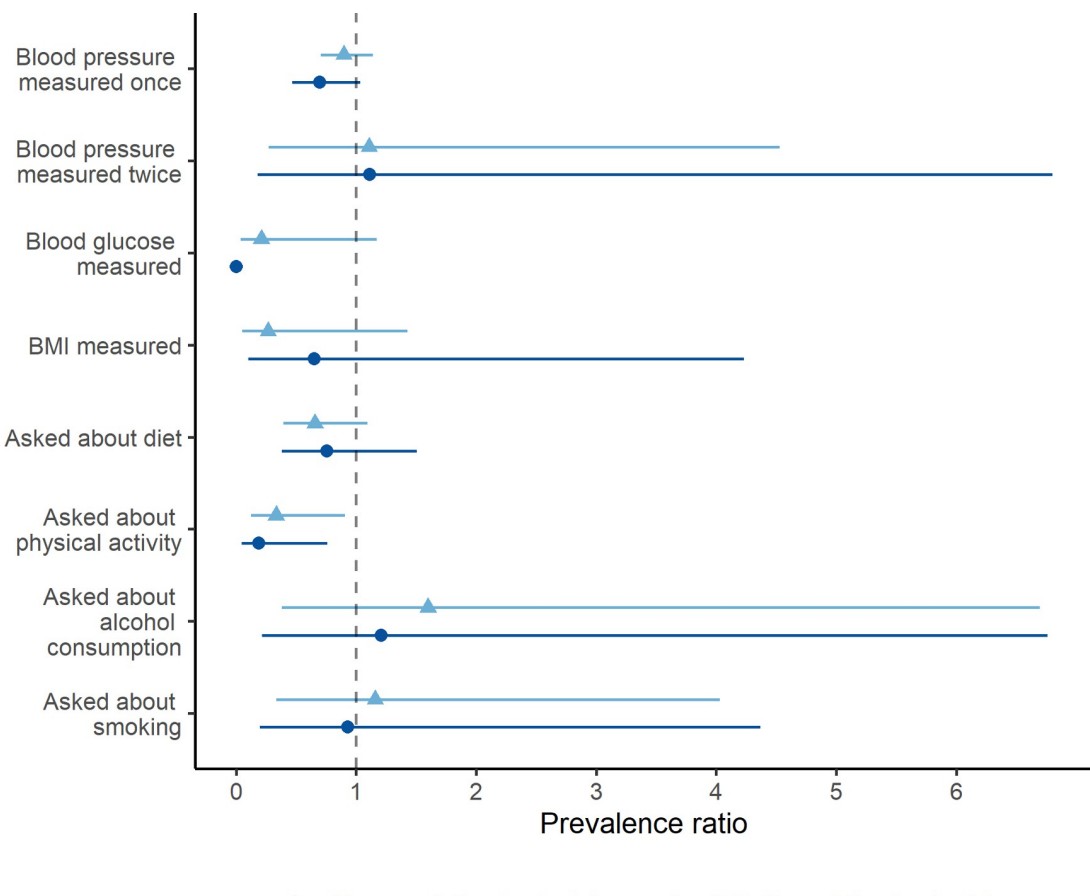

**Fig 2. Association between clinician cadre and CVD care actions based on univariate Poisson regression.** Coefficients are presented as prevalence ratios relative to physicians. Error bars represent 95% confidence intervals. There is no 95% confidence interval for the coefficient on midwives measuring blood glucose since no midwives in the sample measured blood glucose (resulting in perfect prediction and a confidence interval of 0 width).

to measure know-do gaps for CVD care in an LMIC context; however, our results are consistent with several studies from LMICs that find large overall and know-do gaps for a range of other clinical areas, including tuberculosis, child diarrhea, and sick child care [10–12, 14].

An important question is whether the low shares of BMI, blood glucose, and blood pressure measurement is due to clinicians not having access to measurement devices. However, every facility in our study reported having a scale and height board and a sphygmomanometer, making this explanation unlikely for BMI and blood pressure. We did find that only 21% of facilities had access to glucometers and test strips and that clinicians in facilities with these materials were 11 percentage points more likely to measure blood glucose although the 95% confidence interval for this estimate was wide and overlapped the null. Importantly, even among facilities with access to blood glucose measurement devices, just 14% of adult patients had their blood glucose screened, indicating a large remaining gap that is not attributable to equipment availability (S1 Text and Table A in S1 Text).

Our results implicate several areas where clinical knowledge improvements are needed. While the care actions we study are all outlined in the official Cambodian care guidelines, there is limited evidence on whether clinicians practicing in health facilities are aware of and have been trained on these guideline actions. Therefore, measuring what share of clinicians

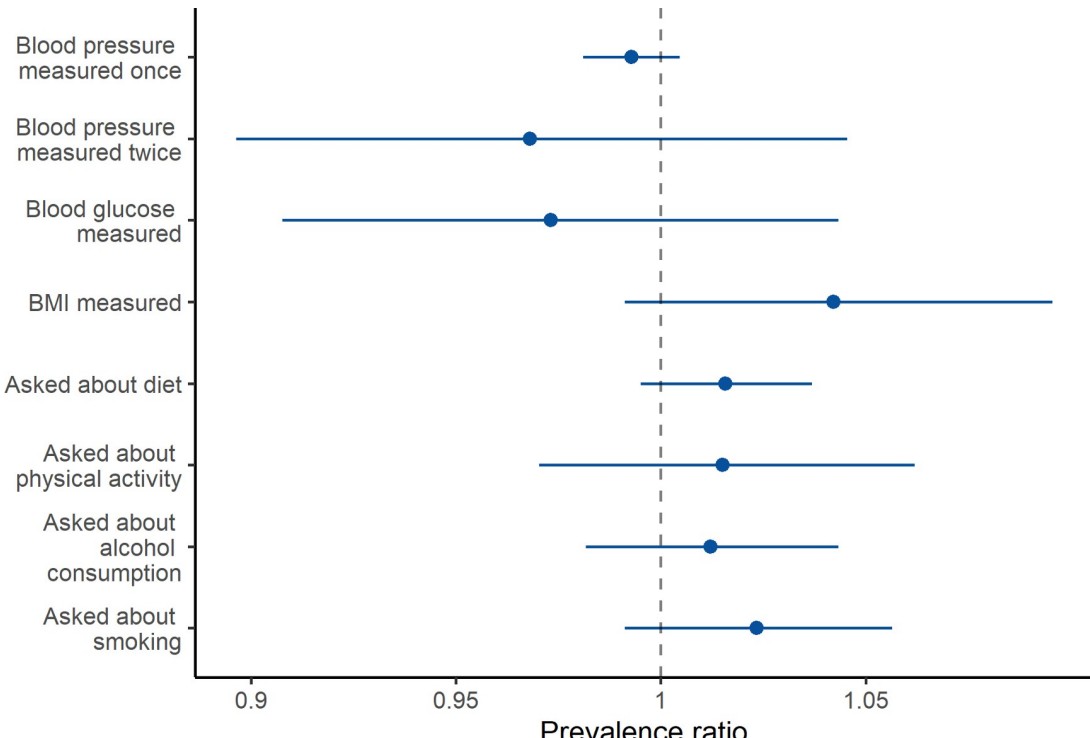

**Fig 3. Association between years practicing and CVD care actions based on univariate Poisson regression.** Coefficients are presented as prevalence ratios associated with a 1-year increase in experience. Error bars represent 95% confidence intervals.

have been trained on guideline actions and subsequently, improving clinician training are likely an important first step for improving clinical care in the country. However, there is mixed evidence on whether education interventions alone are sufficient for meaningfully changing clinician behavior [22–25]. Reconciling these differences and understanding under which circumstances educational interventions are effective will be essential for improving outcomes in Cambodia and similar contexts. Further, our results suggest that even with effective education interventions, there is likely to be a gap between clinician knowledge and behavior. This discrepancy may be related to several factors. For example, clinicians working in public facilities in Cambodia do not have a direct financial—or extrinsic—incentive for providing proper CVD care. Intrinsically, clinicians may not be motivated or find personal satisfaction in providing CVD care if patients do not expect CVD care and thus do not value it in the same way that they value care for acute conditions. Developing and testing strategies that leverage both extrinsic and intrinsic motivators will be crucial for improving clinician behavior [26]. Clinicians may also be habituated to providing acute care and may require significant attention and focus to change their usual behavior towards CVD care. Therefore, clinicians who know and even intend on providing CVD care may face effort barriers and thus opt for the forms of care that they are most accustomed to providing. In such circumstances, providing clinical support tools and checklists that ease the attention and effort needed for clinicians to provide new forms of care may help to improve clinician CVD care behavior [27]. Clinicians may also simply lack the equipment (glucometers) needed to correctly conduct CVD risk assessments.

One of the primary limitations of our paper is that it is based on a non-random selection of public health facilities from 20 out of Cambodia's 94 operational districts. Based on discussions

with Government officials and implementers, the 20 operational districts that were selected to receive the intervention are more likely to have the capacity to provide diabetes and hypertension care compared to the remaining 74 districts. Within districts, the Government selected health facilities with higher patient caseloads and better baseline levels of care quality compared to non-selected facilities. This non-random selection, however, would imply that clinical care behavior among a representative set of facilities would be potentially worse than the levels we document here. Our measures of clinician behavior are based on direct observations and may be biased if clinicians change their behavior when they know they are being watched. This bias, however, would also imply that the already low levels of CVD care behavior observed here may be an overestimate. Our study design only allowed us to measure CVD care at the screening stage and did not allow us to investigate CVD management behaviors by clinicians. Future studies that leverage longitudinal tracking of patients will be essential for measuring care gaps among individuals who have already screened positive for CVD care needs. While we measure many key aspects of preventive care, our analysis does not examine all actions, including for example an assessment of family history. However, as family history is not a substitute for the other actions, this omission would not bias our assessment of the share of clinicians that are not meeting the other clinical actions. Lastly, our study did not investigate whether some types of patients were more likely to receive screening actions compared to others. This is an important question and future research should collect more detailed information on patient characteristics to determine whether certain population groups are being disproportionately overlooked by clinician screening.

Clinician behavior forms a key component of effective CVD prevention care. Yet, there have been few studies that have examined to what extent clinicians in an LMIC context are providing high-quality CVD care and whether care gaps reflect knowledge or other behavioral barriers. Our results reveal large care gaps for key CVD prevention actions, such as measuring blood sugar and blood pressure. Developing interventions that both improve clinician knowledge and address the barriers between knowledge and action will be crucial for closing care gaps and improving CVD prevention in Cambodia and similar contexts.

## Supporting information

**S1 Text. Relationship between equipment availability and blood glucose screening.**
(DOCX)

**S2 Text. Detailed regression results for main study figures.**
(DOCX)

**S1 Checklist. Inclusivity in global research checklist.**
(DOCX)

## Author Contributions

**Conceptualization:** Nikkil Sudharsanan, Matthias Nachtnebel, Chhun Loun, Hero Kol, Till Bärnighausen.

**Data curation:** Sarah Wetzel.

**Formal analysis:** Nikkil Sudharsanan, Sarah Wetzel.

**Funding acquisition:** Nikkil Sudharsanan, Matthias Nachtnebel, Till Bärnighausen.

**Investigation:** Nikkil Sudharsanan, Chhun Loun, Maly Phy, Hero Kol.

**Methodology:** Nikkil Sudharsanan, Chhun Loun, Maly Phy, Hero Kol, Till Bärnighausen.

**Project administration:** Nikkil Sudharsanan, Matthias Nachtnebel, Chhun Loun, Hero Kol.

**Supervision:** Nikkil Sudharsanan, Till Bärnighausen.

**Visualization:** Sarah Wetzel.

**Writing – original draft:** Nikkil Sudharsanan.

**Writing – review & editing:** Nikkil Sudharsanan, Sarah Wetzel, Matthias Nachtnebel, Chhun Loun, Maly Phy, Hero Kol.

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
