## [Decision Letter · Decision Letter 0]

28 Feb 2022

PGPH-D-21-01120

Know-do gaps for cardiovascular disease care in Cambodia: Evidence on clinician knowledge, behavior, and adherence to evidence-based prevention guidelines

Dear Dr. Sudharsanan,

Thank you for submitting your manuscript to PLOS Global Public Health. After careful consideration, we feel that it has merit but does not fully meet PLOS Global Public Health’s publication criteria as it currently stands. Therefore, we invite you to submit a revised version of the manuscript that addresses the points raised during the review process.

Please submit your revised manuscript by  28th March 2022. If you will need more time than this to complete your revisions, please reply to this message or contact the journal office at globalpubhealth@plos.org. Please include the following items when submitting your revised manuscript:

We look forward to receiving your revised manuscript.

Kind regards,

Roopa Shivashankar, MD, MSc

Academic Editor

Journal Requirements:

1. Please amend your current ethics statement to address the following concerns: Please explain how you recorded/documented participant consent, and if the ethics committees/IRBs approved this consent procedure.

3. Please update the completed 'Competing Interests' statement, including any COIs declared by your co-authors. If you have no competing interests to declare, please state "The authors have declared that no competing interests exist".

4. In the online submission form, you indicated that your data will be submitted to a repository upon acceptance.  We strongly recommend all authors deposit their data before acceptance, as the process can be lengthy and hold up publication timelines. Please note that, though access restrictions are acceptable now, your entire data will need to be made freely accessible if your manuscript is accepted for publication. This policy applies to all data except where public deposition would breach compliance with the protocol approved by your research ethics board. If you are unable to adhere to our open data policy, please kindly revise your statement to explain your reasoning and we will seek the editor's input on an exemption. Please be assured that, once you have provided your new statement, the assessment of your exemption will not hold up the peer review process.

5. Please amend your detailed Financial Disclosure statement. This is published with the article, therefore should be completed in full sentences and contain the exact wording you wish to be published.

ii). State the initials, alongside each funding source, of each author to receive each grant.

Additional Editor Comments (if provided):

Reviewers' comments:

Reviewer's Responses to Questions

**Comments to the Author**

1. Does this manuscript meet PLOS Global Public Health’s publication criteria? Is the manuscript technically sound, and do the data support the conclusions? The manuscript must describe methodologically and ethically rigorous research with conclusions that are appropriately drawn based on the data presented.

Reviewer #1: Partly

Reviewer #2: Yes

2. Has the statistical analysis been performed appropriately and rigorously?

Reviewer #1: Yes

Reviewer #2: Yes

3. Have the authors made all data underlying the findings in their manuscript fully available (please refer to the Data Availability Statement at the start of the manuscript PDF file)?

Reviewer #1: No

Reviewer #2: Yes

4. Is the manuscript presented in an intelligible fashion and written in standard English?

Reviewer #1: Yes

Reviewer #2: Yes

5. Review Comments to the Author

Reviewer #1: Major

1. Study population: The title, abstract and introduction mentions “Clinician” as the study population. However, the result section mentions only 10% of the case vignettes and patients observations are really from clinician. The authors should not misguide the readers in this regard. There is no mention about the basic demographic and clinical details of the patients who have been observed as part of consultations. No details available on how many different types of clinical vignettes were used.

2. Further, related to “Adherence to evidence-based prevention guidelines”, neither it was not defined by the authors nor it was comprehensive assessment. Like a) whether history of pre-existing diabetes and hypertension and its treatment status was asked by the doctor, nurse or midwife; and presence of any previous episode of CVD are missing.

3. I could not understand and also not agreeing with the authors on comparing the midwifes and nurses with doctors who have different knowledge and skill levels and also care responsibilities. In this regard, the methodology did not mention

a) what services are provided routinely in the study health centres of Cambodia;

b) what are the brief job responsibilities of midwifes, nurses and doctors, e.g if Midwife or nurse is only responsible for providing counselling service, then of course she will not test the patient for blood sugar;

c) availability of blood sugar machine, sphygmomanometer, stadiometer and weighing machine;

d) the overall and DM/HTN/CVD patient load, the % of acute and non-acute visits and ;

e) availability of any national or local guidelines on prevention and control of CVDs, its implementation and training status;

f) What was the flow of observed patients in the study health centre. Whether all patients goes through everyone or meets only certain people. It is also not clear whether the same patient consultation was observed in all three places and counted as one or three?

g) Whether the care is Paid or unpaid or free for patient?

4. Unfortunately, the authors are also made discussion points on information which is not part of their results. Like Line number 191-192 of page 11, “First, they suggest that individuals with diabetes are being overlooked and left undiagnosed by the public primary care system in Cambodia.”. In the current manuscript, other than testing for blood sugar, the study did not explore anything at all to bring this point for discussion. Similarly, without testing any educational intervention, the authors are discussing its effectiveness.

Minor:

Line 206: Angina is part of CVD and hence it cannot be counted in other clinical areas.

Reviewer #2: Appreciate the authors for choosing an important, relevant topic, the missed opportunities for screening for cardiovascular risk factors in resource limited primary health care setting. The article is well written and the study was conducted following appropriate methodology and tools. This will be interesting for the readers and managers of health programmes. To enrich the article the following may be considered:

1. Section: Abstract: line 28 mentions: “only 2.3% (0.9%, 5.6%) their BMI measured” Instead of “BMI” it should be Blood Glucose.

2. The year of conduction of the study and period of data collection was not mentioned.

3. Two different age groups (>30 and =40 years) were selected for mentioned data collection tools (direct observation and responses following vignettes). The authors may clarify the reasons for having separate "age" criteria and if this age group selection ( if all observed patients were relatively younger) could have affected the result of the study.

4. Section: Statistical Analusis: line 135: “We examined the correlation with years of experience using similar regression models…” The word corelation may be replaced with association

6. PLOS authors have the option to publish the peer review history of their article (what does this mean?). If published, this will include your full peer review and any attached files.

**Do you want your identity to be public for this peer review?** For information about this choice, including consent withdrawal, please see our Privacy Policy.

Reviewer #1: No

Reviewer #2: **Yes: **somdatta patra

---

## [Decision Letter · Decision Letter 1]

24 May 2022

PGPH-D-21-01120R1

Know-do gaps for cardiovascular disease care in Cambodia: Evidence on clinician knowledge and delivery of evidence-based prevention actions

Dear Dr. Sudharsanan,

Thank you for submitting your manuscript to PLOS Global Public Health. After careful consideration, we feel that it has merit but does not fully meet PLOS Global Public Health’s publication criteria as it currently stands. Therefore, we invite you to submit a revised version of the manuscript that addresses the points raised during the review process.

We look forward to receiving your revised manuscript.

Kind regards,

Jasper Tromp

Academic Editor

Journal Requirements:

1. Please provide an Author Summary. This should appear in your manuscript between the Abstract (if applicable) and the Introduction, and should be 150–200 words long. The aim should be to make your findings accessible to a wide audience that includes both scientists and non-scientists. Sample summaries can be found on our website under Submission Guidelines: https://journals.plos.org/globalpublichealth/s/submission-guidelines#loc-parts-of-a-submission

Alternative link: http://journals.plos.org/ploscompbiol/s/submission-guidelines#loc-author-summary

Additional Editor Comments (if provided):

1. Table 1. The colum title (Vignettes and patient observations) not matching with row title (Years practicing).

2. Results: Paragrah 1: Patient observation precentages (by nurses)need to be changed.

3. Though prevalence ratio is correctly mentioned in figures, the same need to be updated in results text (instead of RR).

Reviewers' comments:

Reviewer's Responses to Questions

**Comments to the Author**

1. If the authors have adequately addressed your comments raised in a previous round of review and you feel that this manuscript is now acceptable for publication, you may indicate that here to bypass the “Comments to the Author” section, enter your conflict of interest statement in the “Confidential to Editor” section, and submit your "Accept" recommendation.

Reviewer #1: All comments have been addressed

Reviewer #2: All comments have been addressed

2. Does this manuscript meet PLOS Global Public Health’s publication criteria? Is the manuscript technically sound, and do the data support the conclusions? The manuscript must describe methodologically and ethically rigorous research with conclusions that are appropriately drawn based on the data presented.

Reviewer #1: Yes

Reviewer #2: Yes

3. Has the statistical analysis been performed appropriately and rigorously?

Reviewer #1: Yes

Reviewer #2: Yes

4. Have the authors made all data underlying the findings in their manuscript fully available (please refer to the Data Availability Statement at the start of the manuscript PDF file)?

Reviewer #1: Yes

Reviewer #2: No

5. Is the manuscript presented in an intelligible fashion and written in standard English?

Reviewer #1: Yes

Reviewer #2: Yes

6. Review Comments to the Author

Reviewer #1: (No Response)

Reviewer #2: the authors have adequately addressed comments raised in a previous round of review.

7. PLOS authors have the option to publish the peer review history of their article (what does this mean?). If published, this will include your full peer review and any attached files.

**Do you want your identity to be public for this peer review?** For information about this choice, including consent withdrawal, please see our Privacy Policy.

Reviewer #1: **Yes: **Kathirvel S

Reviewer #2: **Yes: **somdatta patra

---

## [Decision Letter · Decision Letter 2]

13 Jul 2022

Know-do gaps for cardiovascular disease care in Cambodia: Evidence on clinician knowledge and delivery of evidence-based prevention actions

PGPH-D-21-01120R2

Dear Dr. Sudharsanan,

We are pleased to inform you that your manuscript 'Know-do gaps for cardiovascular disease care in Cambodia: Evidence on clinician knowledge and delivery of evidence-based prevention actions' has been provisionally accepted for publication in PLOS Global Public Health.

Best regards,

Jasper Tromp

Academic Editor

Reviewer Comments (if any, and for reference):

Reviewer's Responses to Questions

**Comments to the Author**

1. If the authors have adequately addressed your comments raised in a previous round of review and you feel that this manuscript is now acceptable for publication, you may indicate that here to bypass the “Comments to the Author” section, enter your conflict of interest statement in the “Confidential to Editor” section, and submit your "Accept" recommendation.

Reviewer #1: All comments have been addressed

2. Does this manuscript meet PLOS Global Public Health’s publication criteria? Is the manuscript technically sound, and do the data support the conclusions? The manuscript must describe methodologically and ethically rigorous research with conclusions that are appropriately drawn based on the data presented.

Reviewer #1: Yes

3. Has the statistical analysis been performed appropriately and rigorously?

Reviewer #1: Yes

4. Have the authors made all data underlying the findings in their manuscript fully available (please refer to the Data Availability Statement at the start of the manuscript PDF file)?

Reviewer #1: (No Response)

5. Is the manuscript presented in an intelligible fashion and written in standard English?

Reviewer #1: Yes

6. Review Comments to the Author

Reviewer #1: None

7. PLOS authors have the option to publish the peer review history of their article (what does this mean?). If published, this will include your full peer review and any attached files.

**Do you want your identity to be public for this peer review?** For information about this choice, including consent withdrawal, please see our Privacy Policy.

Reviewer #1: **Yes: **Kathirvel S
